# Experiences of Sexuality in HIV Serodiscordant Gay Couples

**DOI:** 10.3390/healthcare13151788

**Published:** 2025-07-23

**Authors:** María Dolores Ruíz-Ramírez, María Dolores Ruíz-Fernández, María del Rosario Ayala-Maqueda, Marcos Camacho-Ávila, Isabel María Fernández-Medina, María Isabel Ventura-Miranda

**Affiliations:** 1IEMAKAIE (Non-Governmental Organisation), 14005 Córdoba, Spain; m.ruizram01@gmail.com; 2Physiotherapy and Medicine, Department of Nursing, University of Almería, 04120 Almeria, Spain; mrf757@ual.es (M.D.R.-F.); isabel_medina@ual.es (I.M.F.-M.); mvm737@ual.es (M.I.V.-M.); 3Facultad de Ciencias de la Salud, Universidad Autónoma de Chile, Providencia 4780000, Chile; 4Clinical Management Unit of Tabernas, 04200 Almeria, Spain; mariar.ayala.sspa@juntadeandalucia.es; 5Delivery Room Service, Santa Lucia Hospital, 30202 Cartagena, Spain

**Keywords:** human immunodeficiency virus, serodiscordant couples, sexuality, Spain

## Abstract

**Background/Objectives**: Human immunodeficiency virus (HIV) has evolved from a fatal disease to a manageable chronic condition. However, stigma persists, affecting the lives and sexuality of HIV-positive people, particularly in the gay population. Research on their sexuality is limited, highlighting the need for studies that address their experiences and needs. The aim of the study is to explore the individuals’ experiences of sexuality in serodiscordant gay couples. **Methods**: A descriptive study was conducted using thematic content analysis. Data collection was carried out through in-depth interviews. Six gay men who have been and/or are in a serodiscordant relationship for at least one year participated in the research. **Results**: Five sub-themes were identified grouped into two main themes as follows: *sexuality: a complex concept accentuated by HIV* and the *impact of serodiscordance on partners*. **Conclusions**: It is essential to promote accurate information and health services tailored to the needs of people living with HIV while fostering gender equity and combating stigma related to HIV and the gay community. Experiencing sexuality in this context is not only possible but can be full and satisfying when adequate resources are available.

## 1. Introduction

The World Health Organization’s (WHO) definition of sexuality (2006) states that it is “a central aspect” of our experiences that includes “sex, gender identities and roles, sexual orientation, eroticism, pleasure, intimacy and reproduction”. Therefore, sexuality is not limited to sexual intercourse and can be experienced through various forms of intimacy, as well as “thoughts, fantasies, desires, beliefs, attitudes, behaviors, practices, roles, and relationships” [1]. Human immunodeficiency virus (HIV) is a retrovirus that attacks CD4 cells, weakening a person’s immune system against external pathogens [2]. The first cases of HIV infection were documented in 1981, when the US Center for Disease Control and Prevention (CDC) reported the occurrence of atypical pneumonia in young gay men in the United States. Subsequently, in 1983, HIV was identified as the causative agent of the condition [3]. The cause of mortality became known as AIDS (acquired immune deficiency syndrome), defined by the World Health Organization (WHO) (2023) as the most advanced stage of HIV infection, characterised by a profound deterioration of the immune system and exposing the patient to increased vulnerability to various diseases [4]. The latest UNAIDS report (2023) indicates that in 2022, there were 39 million people living with HIV worldwide, with 1.3 million new infections that year. In Spain, “the rate of new HIV diagnoses in 2022 is estimated to be 7.71 per 100,000 inhabitants” [5]. Today, mortality and incidence rates have decreased significantly, and HIV is now classified as a chronic condition [6].

The HIV epidemic has evolved significantly since its emergence in the 1980s, with improvements in the quality of life of infected people through various early detection and prevention strategies, including antiretroviral treatment [7]. The main goal is to achieve an undetectable viral load, i.e., a viral load so low that it cannot be detected by a specific test. In this way, the possibility of transmission of infection is zero [8]. The possibility of reaching undetectable status has been a revolution [9]. The motto undetectable equals untransmissible was defined after the European PARTNER-2 macro study involving 14 countries that followed 1000 serodiscordant same-sex couples for 8 years [10]. The results of this study showed that stable couples, whose HIV+ partners took antiretroviral treatment under medical supervision, did not become newly infected after unprotected sex [11].

In addition, it is important to note that the risk of infection has significantly decreased in recent years due to increased access to early HIV detection tests [12] as well as new drug combinations. On the one hand, pre-exposure prophylaxis (PrEP), an antiretroviral treatment given to HIV-negative people who are highly vulnerable to the virus, reduces the risk of transmission by up to 99% [13]. On the other hand, post-exposure prophylaxis (PEP), a combination of drugs to be administered after possible HIV risk within 72 h of exposure for a duration of 28 days, reduces the risk of transmission by up to 99.2% [14].

However, being a person diagnosed with HIV continues to be a risk factor or a greater vulnerability due to the great stigma attached to this reality, which is based on the perception that this infection represents a threat both to individuals and to society as a whole [3]. Discrimination against people living with HIV continues to be reported, resulting in human rights violations [11]. This is supported by the European macro-survey entitled “HIV is sorted?”, which was carried out in 2018 and involved 24,000 people from twelve countries. The results showed a high level of ignorance about HIV protection, prevention, and transmission [15].

In Spain, a Social Pact for Non-Discrimination and Equal Treatment Associated with HIV has been created because although legislation protects the rights of these people, the discrimination that accompanies infection is still present [16]. The existence of stereotypes about certain diseases, especially sexually transmitted diseases, and the dissemination of incorrect information about how HIV is transmitted leads to discrimination in terms of access to social, legal, and health care services, as well as in the labour market and in obtaining housing [17]. Stigma has a negative impact on prevention, diagnosis, treatment, and quality of life, often resulting in a negative and distorted self-image due to fear of judgement and/or rejection by people when they learn their HIV status [18]. Stigma can also be transferred to different areas of life, especially in interpersonal relationships, a turning point being in relationships with partners, where it can affect the vital development of the person with HIV, especially if their partner does not have the virus [19]. A recent study in Spain found that 32.2% of people with HIV surveyed experienced rejection by their sexual partners [17].

In this context, the term serodiscordant refers to couples when one partner is HIV-positive and the other is HIV-negative [20]. A diagnosis of HIV can limit sexuality, significantly affecting the psychosocial wellbeing [12] and sexual health [21] of the individuals involved. Serodiscordant couples, especially those in stable relationships, face the significant challenge of HIV transmission risk influenced by factors such as sexual behaviour, viral load, and the presence of other sexually transmitted infections [22]. A relevant aspect is the differentiated vulnerability between genders and between sexual orientation. Several studies indicate that women occupy a more vulnerable position [23,24,25] along with gay men, as the population has the myth that HIV infection affects gay men, generating rejection, discomfort, and promoting homophobia [26] so that feelings of rejection and discrimination are more accentuated in HIV-positive gay men [27]. For a long time, the sexual rights of people living with HIV have been neglected, and receiving a diagnosis could be interpreted as the end of sexual life [28]. However, it has been shown that being HIV+ is not a barrier to enjoying a full sexual life [9]. Sexuality is an essential dimension and is understood as a multidimensional phenomenon that influences people’s individual and social lives [29].

This study addresses a significant gap in the literature regarding the sexuality and lived experiences of serodiscordant gay couples. It is essential to explore their experiences and identify their needs in order to uphold their right to accurate, evidence-based information on sexuality and sexual health [30]. The main research question is what are the sexual experiences of serodiscordant individuals and which factors related to HIV infection differentiate these experiences? Accordingly, the aim of this study is to explore the experiences of serodiscordant gay couples in relation to their sexuality.

## 2. Materials and Methods

### 2.1. Design

This is a descriptive qualitative study based on thematic content analysis [31]. This design follows a naturalistic research paradigm, which seeks to understand phenomena from the perspective of those who experience them.

### 2.2. Participants and Setting

This study was conducted in Andalucia (Spain). Participants were selected using snowball sampling due to the difficulty of accessing the target population. We contacted testing centres or clinics for HIV+ individuals and LGTBI association. Recruitment was facilitated through referrals from other participants who met the inclusion criteria, which were being or having been in a serodiscordant relationship for at least one year, being over 18 years of age, not being in the immediate period following an HIV diagnosis and/or grieving process, and providing informed consent. Exclusion criteria included couples in which both partners were HIV-positive, having been in a serodiscordant relationship for less than one year, and individuals undergoing a complex therapeutic process following HIV diagnosis.

An email was sent to 46 men. However, only 6 decided to participate. In total, 28 declined to participate because they were afraid to discuss this topic, 6 did not want to participate because they did not feel comfortable talking about this topic, and 6 did not respond to the email. The sample size did not reach information saturation because the sample was small but rather was determined by the number of men who decided to participate. Sociodemographic data are presented in Table 1.

### 2.3. Data Collection

Data collection took place between November and December 2024 via open-ended in-depth interviews. Participants were recruited via an email invitation, in which the conditions and objectives of the study were explained. Subsequently, an appointment was made with the individuals who gave consent to participate in the study.

In order to ensure natural and open-ended interviews in which the participants felt they could express themselves openly, the researchers received training and practised the interview techniques before conducting the interviews.

An interview protocol was used, providing information about the objective, ethical issues, consent, and questions to guide the conversation (Table 2). The average duration of the interviews was 60 min. The interviews were recorded on a digital device with prior consent from the interviewee. They were transcribed immediately after taking place so that they could be revised and enhanced with the researcher’s notes.

### 2.4. Data Analysis

ATLAS.ti software 25.0.0 (ATLAS.ti Scientific Software Development GmbH, Berlin, Ger- many) was used for data analysis, integrating the transcriptions, the coding and cate-gorization system, and memos in a project. Thematic analysis was carried out following an inductive strategy (the themes are not based on a literary review but instead emerge from the data), in compliance with the steps described by Mayring (2000) [31] and adapted for psychoeducational and health research [32,33].

First step: Selecting the object of analysis within a communication framework. The research group started by defining their theoretical, professional, or scientific stance [34].

Second step: Preliminary analysis. In this stage, the transcripts were read with the aim of gaining a general understanding and holistic vision of the texts.

Third step: Defining the units of analysis. Through a careful re-reading of each document, significant sentences or extracts were chosen as units of analysis. We aimed to select units of analysis sufficiently large to be considered as a whole but small enough to be a relevant meaning unit during the analysis process [32]. A total of 205 quotes were selected as units of analysis and were codified in the subsequent phase.

Fourth step: Codification, namely establishing rules of analysis and classification codes. The “open coding” function of ATLAS.ti software was used to assign codes to the units of analysis, giving them meaning. The “insert comment code” function was applied to define the usage rules of each code as well as the requirements for the successive codification of units of analysis with each code.

Fifth step: Categorization. The codes were analysed semantically to explore their meaning and group them into themes and subthemes, using the ATLAS.ti function of “group codes” and “link codes”. In order to define the emerging themes, conceptual and explanatory memos were created.

Sixth step: Final integration of the findings. In qualitative analysis, the wording and revision of the conclusions are analysed. In our final report, we used themes, memos, and codes, allowing us to achieve robust results with the support of all previously carried out work.

### 2.5. Rigour

Reliability was achieved through a researcher’s triangulation throughout the process; various researchers revised the entire coding and categorisation system, discarding the themes and subthemes for which there was not a general consensus. To increase the internal validity, all opinions and experiences were represented, and the transcripts and analysis were returned to some participants for them to confirm the content and the researchers’ interpretations. To improve the trustworthiness of the study, a checklist was used that includes different phases of preparation, organisation, and reporting of the study [32].

### 2.6. Ethical Issues

This study was approved by the University of Almeria’s Ethical and Research Committee (EFM 357.24). The core aspects of sexuality according to the WHO were taken into account [1]. All participants were informed of the objective of the study, they participated voluntarily, and they gave written consent to participate in the study. They had the possibility to end the interview at any given moment. In order to ensure anonymity and confidentiality, the participants were identified using an alphanumeric code. All data have been handled in accordance with European and Spanish data protection laws.

## 3. Results

After transcription and analysis of the data, 21 codes emerged, and these were grouped into five sub-themes. In turn, two main themes emerged from these sub-themes (see Table 3).

### 3.1. Sexuality: A Complex Concept Emphasised by HIV

Sexuality is a fundamental aspect of an individual’s life and a complex concept encompassing multiple dimensions of human existence. However, the perception and experience of sexuality varies considerably among people, reflecting the subjective and individual nature of sexuality. This complexity is even greater for people living with HIV, as the condition influences their daily lives, including their own sexuality and that of their partner(s).

#### 3.1.1. Concept and Experience of Sexuality: “It Is Very Broad and Personal”

Participants in the study agreed that sexuality is a complex and inherent human phenomenon, encompassing multiple aspects of the construction of identity.

*For me it is a very broad term, which not only focuses on sexual practices, it focuses on care, how I live my sexuality, how I express myself, how I eroticise myself and how I eroticise others. How I take care of myself. And also care for the other person or persons*.(P-5)

On the other hand, participants noted that their experience of sexuality has evolved over time, influenced both by the sexuality education they have received and the experiences they have had. This reflects the impact of socio-cultural context on sexuality, depending on the environment that surrounds each individual. Most agreed that, being part of the LGTBIQA+ group, coming to terms with their sexuality during childhood and adolescence was a complicated process.

*It is a complicated path, I am gay, in a small town, where the search for resources or socialisation groups, like me, was more complex, as I had a very strong homophobia where until I was 24 I didn’t even dare to tell someone that I liked boys*.(P-4)

*On the basis that, being gay, already, that implies that you have lived a part of your sexuality totally hidden, and besides, it has always been heard that AIDS is for queers*.(P-6)

#### 3.1.2. Fear of Diagnosis

There exists two perspectives—that of the HIV-positive person at the time of diagnosis and that of their partner. On the one hand, HIV-positive individuals consistently reported experiencing their diagnosis as a devastating event, characterised by intense fears and a perception that their life was in immediate danger.

*I found out in the hospital because of a health problem, they tested me and realised I was in the AIDS stage. So it was a traumatic, devastating diagnosis that left me in shock and I didn’t know how to assimilate*.(P-4)

*He told me that I was positive, he asked me some questions, I didn’t say anything because I was in a state of shock and that’s where I went with that little information and that bad management. And at the beginning it was very complicated, I had a lot of stigma. I panicked that someone would find out*.(P-5)

On the other hand, we found two different views among the couples, some of whom immediately normalised their knowledge of the diagnosis, while others panicked because of their lack of knowledge about HIV infection.

*I thought it didn’t even exist, I started to get tested, I thought I also had HIV, which in the end didn’t come out positive, but well, I was able to get tested more than seven to eight times a year and believing, and I felt that I had HIV, so it affected my relationship. I tried it with him, but in the end it didn’t come out*.

*I remember my partner even asked me if I didn’t mind if he had HIV. You notice that your partner is afraid that you will leave him or she is afraid of infecting you, but at the end of the day you have another characteristic like any other person, like people with disabilities or whatever*.(P-3)

### 3.2. Impact of Serodiscordance on Couple Dynamics

An HIV diagnosis inevitably has an impact on the life of an HIV-positive person that not only affects their physical health but also encompasses their psychosocial wellbeing. This impact extends to interpersonal relationships and can lead to environments of isolation and discrimination. Acceptance of the diagnosis is a lengthy and complex process, fraught with challenges, and it is a crucial factor in building a serodiscordant relationship, as it can represent a significant turning point in a couple’s relationship.

#### 3.2.1. Strategies for Coping with Change After Diagnosis

Living with HIV involves adapting to significant changes in daily routines and daily life, coping with difficult conversations with family, friends, and partners, managing medical information overload, and dealing with an emotional process that can be overwhelming. In addition, it is common to have to deal with the social stigma associated with the virus, which adds an additional layer of psychological stress, and a variety of coping strategies are vital. Participants in this study mentioned professional medical and psychological support, HIV education received, and support from friends, family, or a partner’s trust and companionship in the process.

*The hospital service is wonderful. It is very reassuring. As far as possible they make it easier for you too, they try to take the heat off the matter, they do it in a relaxed way and that makes things a little bit easier for you*.(P-2)

*I was afraid of rejection from my friends, my parents and my partner, I thought that no one else in the world would want to be with me and I was afraid of having sex in case I infected someone, having to verbalise all this was definitely the hardest part*.(P-4)

#### 3.2.2. Negative and Positive Aspects of Serodiscordance in Couples

A common theme among participants is that serodiscordance had a significant negative burden on the partner and their relationship with their partner. Fear, fear of rejection, and fear of contagion in relationships and not being able to fully enjoy themselves is something that tormented them.

*Many times I thought it would be easier if we were both positive, I was very afraid of infecting him and at the beginning I didn’t enjoy sex at all and although he thought it was true I don’t think it was*.(P-3)

*I became paranoid about sex, I had an irrational fear of being infected, of the condom breaking, I didn’t enjoy it at all*.(P-4)

Participants identified several positive aspects of being in a serodiscordant relationship. These relationships often foster a higher level of trust and more open communication, as talking about HIV health and prevention becomes essential for both parties. This can strengthen the emotional bond and reduce fear around the virus, reinforcing their support for each other and enabling them to face challenges together.

*This allowed me to become more informed, to prioritise my sexual health, how I can reduce risks in my relationships, and so that helped me. It also helped me to become more informed about the infection and to avoid that discrimination, that stigma that I had around HIV. It helped me to have safe relationships and enjoy sex safely. If I could be with a person with HIV again*.(P-1)

*When you are in a couple, you not only care about your health, you care about your partner’s health, and that makes the relationship stronger and creates stronger bonds both in sex and in day-to-day life*.(P-2)

#### 3.2.3. Perception of Stigma in Everyday Life

HIV-related stigma and discrimination are significant barriers for people living with HIV, contributing to social isolation, non-adherence to treatment, and lack of access to appropriate medical services. These phenomena are based on prejudice, misinformation, and unfounded fear, leading to a range of negative consequences in the lives of those affected. HIV-related stigma, despite medical and educational advances, remains a major obstacle. All participants agreed that they had experienced various forms of discrimination related to their HIV status. These ranged from refusal to date or have intimate relationships, refusal of a medical facility to allow their serodiscordant partner to donate blood, to exclusion by family members. These episodes are not only hard but also have a profound emotional impact, affecting self-esteem and mental health.

*My mother didn’t like me being with someone with HIV, because to her it was something that might affect me or I might get infected, which was very upsetting for me*.(P-3)

*People who tell you I don’t sleep with anyone who has a virus, I know you are undetectable, but I don’t want to meet you, because you haven’t told me before*.(P-4)

## 4. Discussion

The aim of this research was to explore the experiences of serodiscordant gay couples regarding sexuality, recognising sexuality as an undeniable aspect of human life and a fundamental right that should be fully exercised regardless of individual characteristics [30]. The conception of sexuality among the participants is broad and encompasses dimensions that transcend sexual practices such as affection, self-care, or couple communication. Furthermore, the presence of HIV does not imply the interruption of sexual life, as expressed by the participants in this study. A macro study that confirms this idea is the well-known PARTNER-2 study that demonstrated that undetectable viral load leads to the elimination of the risk of HIV transmission in unprotected relationships, ensuring quality of life among the nearly 1000 couples participating in the study. Rather, it requires adaptation to a new reality that allows for a full and satisfying sexuality [11].

Regarding the impact of an HIV diagnosis, participants reported experiencing shock, fear, and disbelief; however, over time, the situation tends to normalise, with a diversity of feelings and disparity in communicating their diagnosis. This is consistent with other research finding this diversity of emotions, feelings, and ways of communicating the diagnosis, which is underpinned by the overriding need to protect oneself [35].

How individuals cope with the diagnosis largely depends on access to information, available social support, and even the presence of coping strategies. It should be noted that disclosure of HIV status to partners and/or close associates is a complex and individual process, as while some people disclose openly, others choose to keep it secret due to fear of rejection. Similar results with other studies showed that more than 40% of people with HIV did not share their HIV status with their partners due to feelings of fear or anxiety about their partner’s reaction [36].

Several studies have found individual differences in the decision to disclose an HIV diagnosis. On the one hand, gender is one of the most important determinants. Gender is a sociocultural concept that refers to the roles, behaviours, activities, and attributes that a society considers appropriate for men and women. Unlike sex, which is based on biological characteristics, gender is a social construct that can vary between cultures and over time [37]. In this regard, gender inequality is highlighted in terms of unequal access to resources, as well as women’s greater exposure to violence and unplanned pregnancies [19]. This same opinion was reflected in another article which highlighted the testimonies of the 20 women interviewed, who underlined how health care services do not take into account their reality as HIV carriers [38]. Unfortunately, it was not possible to establish contact with women in serodiscordant relationships, which highlights the difficulty of access to this group and reinforces the structural stigma that these women face, limiting their participation in the research. On the other hand, sexual orientation and the stigma of having HIV and being a gay man encompass a different set of struggles, since the origins of HIV are in the gay community in the USA. Some of the participants reported that they were aware of the double stigma of being a gay man and HIV-positive. Several studies on the subject note this stigma in the gay community, which accentuates feelings of fear and rejection [39], resulting in difficulty in relating, maintaining relationships, and therefore enjoying sexual health [40,41].

Across the board in this study, participants reported their own coping strategies upon learning of their diagnosis, such as turning to qualified health professionals, supportive family and friends, and NGOs that have provided them with HIV education in addition to self-education. This has shown that individual coping strategies are vital in coping with the diagnosis, as well as the disclosure of HIV status to a partner. This is reinforced in the study by Millogo et al. (2019), in which HIV-positive people who had a negative perception of their infection did not feel confident sharing their HIV status because of the various social repercussions compared to those who had more information about their infection, who were encouraged to disclose [40]. Similarly, studies have shown, with limitations, that the implementation of an educational programme aimed at this group significantly reduced risky sexual behaviour and improved sexual relations [41].

Disclosing HIV status is a major determinant in the lives of people with HIV, affecting their mental health, interpersonal dynamics, and adherence to antiretroviral treatment. Disclosure of the diagnosis can have adverse repercussions that affect their wellbeing, in some cases leading to social isolation and deterioration in their quality of life [35]. However, in this research, not only negative aspects emerged in terms of disclosure of diagnosis and serodiscordance with their partner. Positive aspects such as increased sexual health awareness, strengthened mutual commitment, and a more informed and less prejudiced attitude towards HIV emerged.

When it comes to sexual relations within the couple, fear and anxiety about infecting the other partner is a major factor, which has a negative impact on sexual relations, as the study participants explained. Similar findings have been reported in other studies, where people living with HIV experience feelings of panic and anxiety about having sex because they become obsessed with infecting their partner [42], and their sexual practices decrease in the early days after diagnosis [43].

With regard to the stigma associated with HIV, it has been highlighted throughout the research process that it is an aspect that continues to be present in the daily lives of people with HIV, including their closest environment, coinciding with the experiences gathered in the study by Farago et al. (2018), in which the six people interviewed stated that they had suffered situations of social rejection in different areas, highlighting the feeling of loneliness that these experiences produce. The persistence of stigma underlines the need to continue promoting education and awareness campaigns to reduce prejudice and promote a more inclusive and evidence-based approach [44]. Also, several pieces of evidence highlight the importance of optimal performance on the part of health professionals in accompanying these couples. The action of previously qualified nursing professionals with educational tools combats prejudice, as well as contributes to the biopsychosocial health of these people [45,46]. It is necessary to highlight that the studies published on this topic are limited, and some are even outdated and tend to focus on dimensions other than sexuality (perception of the risk of transmission, use of PrEP, adherence to antiretroviral treatment, etc.).

## 5. Conclusions

Sexuality is an essential component of people’s lives, regardless of their HIV status. On the one hand, the sexuality of people living with HIV, as well as their respective partners, is mainly influenced by the stigma surrounding HIV. Their sexual life has to undergo some adaptations to the new reality but can be fully exercised. On the other hand, knowing the HIV status of partners has a significant impact, and access to appropriate resources contributes to the normalisation of this reality, even leading to individual empowerment.

Challenges remain around the disclosure of HIV status, which remains a complex process deeply influenced by personal, social, and structural factors. The role of the context of the HIV-positive person is highlighted, and partner communication in dealing with difficulties is key and essential in that relationship and the couple’s sexuality.

The stigma associated with HIV and the gay community is still present in society and needs to be made visible. Experiences of discrimination persist in different settings, including the family environment and health care, which reinforces the need to continue promoting awareness-raising initiatives aimed at both the general population and health professionals. One of the key tools for this is comprehensive sexuality education that perpetuates safe practices and works towards the respect and acceptance of diversity.

It is vital to promote accurate information and health services tailored to the needs of people living with HIV, supporting gender equity and the elimination of stigma. Experiencing sexuality in this context is not only possible but can also be fulfilling and satisfying when adequate resources are available.

## Figures and Tables

**Table 1 healthcare-13-01788-t001:** Sociodemographic data of participants.

Participant	Sex	Age	HIV Serological Status	Sexual Orientation	Nationality
1	M *	33	negative	Homosexual	Spain
2	M *	28	positive	Homosexual	Spain
3	M *	27	negative	Homosexual	Spain
4	M *	30	positive	Homosexual	Spain
5	M *	41	positive	Homosexual	Spain
6	M *	37	negative	Homosexual	Spain

* M = man.

**Table 2 healthcare-13-01788-t002:** Interview protocol.

Phase	Title	Content/Example of Questions
Introduction	Objective of the research	To learn about the sexual experiences of serodiscordant couples.
Home	How would you define your relationship as a couple? How would you define your sexuality? How did you discover that one of you was HIV-positive and the other was not? How did this revelation impact your relationship and your sex life?	
Development	3. What kind of conversations have you had about serodiscordance and its relationship to sexuality? How have you addressed fears or concerns related to HIV transmission? 4. Have you experienced changes in the frequency or nature of your sexual relations due to serodiscordance? How have you managed these changes as a couple? 5. How have the experiences of serodiscordance affected your perception of sexuality and the relationship in general? Have you found positive aspects or new ways of connecting as a couple? 6. Have you experienced stigma or discrimination in your sexual life due to serodiscordance? How have you dealt with these situations? 7. Did you have any mistaken beliefs about infection before you were with your partner? 8. Did you find any supportive figures in your close circle? 9. Have you sought professional counselling or support in relation to your sex life and serodiscordance? To what extent has this support helped you?	
Closing	Final question	Is there anything else you would like to say on the subject?
	Acknowledgements	We thank you for your time. Remind him that his testimony will be of great help.

**Table 3 healthcare-13-01788-t003:** Themes, sub-themes, and units of meaning of analysis.

Theme	Sub-Theme	Codes
Sexuality: A complex concept accentuated by HIV	Concept and experience of sexuality: “it is very broad and personal”	Complex; biopsychosocial; culture; affective relationships; pleasure; sex; sexual relations
	The fear of diagnosis	Social isolation; fear
Impact of serodiscordance on the couple	Strategies for coping with change after diagnosis	Effective communication; HIV education; professional support; supporting friends; trust
	Negative and positive aspects of serodiscordance in couples	Sexual health; paradigm shift; sexuality education
	Perception of stigma in everyday life	Irrational fears; family rejection; rejection by partners; low self-esteem

## Data Availability

The data in this study comprise audio recordings and their transcripts (confidentially), which are stored in an ATLAS.ti software project.

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
