# Peer review of "Experiences of Sexuality in HIV Serodiscordant Gay Couples"

_healthcare, 2025, doi:10.3390/healthcare13151788_

Round 1

Reviewer 1 Report

Comments and Suggestions for Authors

The subject covered in this article is hardly ever discussed, even though from the perspective of the quality of life of HIV-positive people and their healthy partners, it is highly relevant. Its significance is particularly evident as regards serodiscordant gay couples, who experience double discrimination and exclusion. Their sexual rights are disregarded. I would like to congratulate the Authors on having the courage to tackle a marginalised issue surrounded by misconceptions and stigma. I found the article and your interpretation very insightful.

Introduction: accurately written, carefully planned, emphasising key issues, based on the latest medical and scientific knowledge;

Sample selection: a relatively small sample; it is recommended that in the future the sample size be increased and a different, more reliable sampling technique than the ‘snowball’ method be applied (e.g. via testing centres or clinics for HIV+ individuals); please specify in the text the number of individuals who were sent an email invitation to participate in the study;

Research method: precise, respectful of ethical principles;

Results and conclusions: presented in a professional manner and truly engaging; I consider them applicable to other European countries; they repeatedly highlight the critical issue of fear and anxiety permeating various aspects of the respondents' lives; I would also suggest putting greater emphasis on the overall importance of sexual activity for both the well-being and quality of relationships of HIV-positive individuals.

Author Response

Thank you very much for taking the time to review this manuscript. We appreciate your thoughtful and generous feedback.

Reviewer 2 Report

Comments and Suggestions for Authors

Thank you for the opportunity to review this manuscript. The last word of the title is misspelled. It read ‘couplestle’ instead of ‘couples.’ However, the title captures the manuscript’s content. The topic is relevant to policymakers, healthcare providers, and the general public. The keywords should include the country where the study was conducted. The abstract is a correct summary of the manuscript. The introduction orients the readers to the topic, identifies research gaps, and states the aim of the study. Some aspects are missing in the materials and methods section. Some new results only appear in the discussion section.

MAJOR REVISIONS

Abstract

  1. Add the data collection tool under the methods section.

Materials and Methods

  1. In section 2.2, the authors should state the region where the study was conducted.
  2. How was the sample size determined?
  3. How many people refused to participate?
  4. Where was the data collected?
  5. Was the interview guide pilot-tested?
  6. Discuss data saturation.
  7. Table 1 should be moved to the results section.

Results

  1. The first section should be ‘Characteristics of participants’ where Table 1 will be presented. Additionally, summarise the characteristics in narrative form.
  2. In line 181, the authors state that 3 main themes emerged, yet only 2 are discussed. This should be corrected.

Discussion

  1. In lines 131-132, the authors state, ‘Across the board in this study, participants have acquired different coping strategies upon learning of their diagnosis, ….’ This is a new finding which was not presented under results. The authors should either present this under results or remove it from the discussion.
  2. In lines 368-369, the authors state, ‘….some testimonies stand out in which it is reflected that they have suffered discriminatory treatment by health professionals, ….’ This finding was not presented under results. The authors should either remove it or add it to the results.

Reference

  1. Ensure that references from the web adhere to journal guidelines.

MINOR REVISIONS

  1. The sentence in lines 58-60 is incomplete. Rephrase for clarity.

Reviewer 3 Report

Comments and Suggestions for Authors

Comments are attached.

Round 2

Reviewer 2 Report

Comments and Suggestions for Authors

The authors have satisfactorily addressed all my comments.